# Relationship between the Dynamics of Flavor Compounds and Microbial Succession in the Natural Fermentation of Zhalajiao, a Popular Traditional Chinese Fermented Chili Paste

**DOI:** 10.3390/foods12203849

**Published:** 2023-10-20

**Authors:** Luhan Huang, Yanyan Tang, Jiong Zheng, Jianquan Kan, Yun Wu, Yating Wu, Sameh Awad, Amel Ibrahim, Muying Du

**Affiliations:** 1College of Food Science, Southwest University, Chongqing 400715, China; 2Chinese-Hungarian Cooperative Research Center for Food Science, Southwest University, Chongqing 400715, China; 3Chongqing Key Laboratory of Speciality Food Co-Built by Sichuan and Chongqing, Chongqing 400715, China; 4Chongqing Houjie Pharmaceutical Group Co., Ltd., Chongqing 404100, China; 5College of Food Science and Pharmaceutical Science, Xinjiang Agricultural University, Urumqi 830052, China; 6Institute of Quality Standards & Testing Technology for Agro-Products, Xinjiang Academy of Agricultural Sciences, Urumqi 830091, China; 7Faculty of Agriculture, Alexandria University, Alexandria 21500, Egypt

**Keywords:** Zhalajiao, physicochemical properties, flavor components, microbial succession, correlation analysis

## Abstract

Zhalajiao, a traditional Chinese fermented food, is popular due to its unique flavor. Traditional Zhalajiao fermentation is closely related to flavor compounds production. However, the mechanisms underlying the formation of these crucial flavor components in Zhalajiao remain unclear. Here, we explored the dynamic changes in physical and chemical properties, microbial diversity, and flavor components of Zhalajiao at various fermentation times. In total, 6 organic acids, 17 amino acids, and 21 key volatile compounds were determined as flavor components. In Zhalajiao, *Lactobacillus* and *Cyanobacterium* were the main bacteria that were involved in the formation of crucial flavor compounds. *Candida* showed a significant correlation with 14 key flavor compounds during fermentation (*p* < 0.05) and was the main fungal genus associated with flavor formation in Zhalajiao. This research offers a theoretical foundation for the flavor regulation and quality assurance of Zhalajiao.

## 1. Introduction

Zhalajiao is a traditional Chinese fermented food with distinctive characteristics that has become increasingly popular among people in the central and south-western regions of China due to its vibrant color, appetizing sourness, and mellow aftertaste [1]. Traditionally, it is produced through spontaneous fermentation by using cereals (mainly corn or rice) and fresh chilies (Xiaomila, Erjingtiao, and Niujiaojiao) as raw materials under the action of naturally occurring microorganisms [2]. The entire fermentation cycle lasts 30–90 days at room temperature. With its characteristic flavors of sourness, aroma, spiciness, and freshness, Zhalajiao can be served as a standalone dish after being fried or used as a spice to improve the taste of other dishes [3]. During fermentation, carbohydrates, lipids, proteins, and other substances in Zhalajiao are decomposed, resulting in the formation of various flavor components, such as organic acids, free amino acids (FAAs), and volatile compounds (esters, acids, terpenes), which have been shown to significantly contribute to the flavor characteristics of fermented Zhalajiao [4]. Additionally, microorganisms are crucial to the creation of flavor components during fermentation. The diversity of microbial composition places fermented foods in a complex micro-ecological environment, leading to the diversity of the flavors of these products. Given the relatively uncontrolled fermentation of Zhalajiao, the presence of various microorganisms in the air, water, and fermentation tanks and the use of various raw materials result in the formation of a microbial community with a complex structure in the final product. In traditional fermented foods, this complex natural microbial community is related to physicochemical properties and affects flavor characteristics [5]. 

Some research has been performed on microbial diversity in Zhalajiao products. Guo et al. [3] used high-throughput sequencing and culture-dependent methods to collect and study corn-based Zhalajiao samples from two regions. They found remarkable discrepancies in the bacterial community composition of corn-based Zhalajiao samples from two regions. *Companilactobacillus*, *Lactiplantibacillus*, and *Corynebacterium* were the common dominant genera in the two samples. Cai et al. [1,2,6] studied the bacterial community and flavor characteristics of rice-based Zhalajiao produced in different regions from differing raw materials. Rice-based Zhalajiao made in different regions from varying ingredients had significantly distinct microbial communities and a sensory quality, with *Lactobacillus* being the predominant bacterial genus. However, the above research merely discussed the bacterial community at the culmination of rice-based Zhalajiao fermentation. In a previous research paper, corn-based Zhalajiao with a fermentation time of 0–60 days was collected to study the dynamic changes in microflora and flavor components during its fermentation [4]. *Lactobacillus* and *Kazachistania* were considered to be the most preponderant bacterial and fungal genera, respectively. Furthermore, *Lactobacillus*, *Acetobacter*, *Kazachistania*, *Wickerhamyces*, *Kluyveromyces*, and *Cyberlindnera* were the major microbes participating in the generate of flavor components. Nevertheless, to our knowledge, no research on the composition and succession of bacteria and fungi in the fermentation of rice-based Zhalajiao is available, and the relationship among the major flavor compounds, physicochemical properties, and associated microorganisms of this product is unclear. Therefore, studying the dynamic succession of physicochemical properties, flavor compounds, and microflora of rice-based Zhalajiao during fermentation and the relationship among these factors is crucial for the production and quality control of rice-based Zhalajiao. 

For this study, the changes in physicochemical indices, organic acids, FAAs, and volatile components during Zhalajiao fermentation were measured. High-throughput sequencing was applied to analyze the combination of and change in microbial communities during Zhalajiao fermentation. In addition, the relationship between flavor components and microflora was explored. The aim of this research is to lay the foundation for the identification of the core functional microbial and improvement in the quality of traditional Zhalajiao.

## 2. Materials and Methods

### 2.1. Materials and Reagents

Mature fresh erjingtiao was bought from the local market (Guizhou, China). Standards, including oxalic, acetic, malic, tartaric, lactic, citric, and succinic acids, were purchased from Solaibao (Beijing, China), and 2-octanol and C7–C40 n-alkanes were purchased from Aladdin (Shanghai, China) and Anpu (Shanghai, China), respectively. Kelon (Chengdu, China) supplied chemicals of analytical purity, including phosphoric acid, potassium dihydrogen phosphate, 5-sulfosalicylic acid, n-hexane, sodium hydroxide, phenolphthalein, concentrated sulfuric acid, anthrone, potassium sodium tartrate, phenol, sodium sulfite, silver nitrate, 3,5-dinitrosalicylic acid, and chromatographic-grade methanol.

### 2.2. Zhalajiao Preparation and Collection

Fresh red chilies were cleaned and cut into pieces and then mixed well with an equal weight of rice flour and 6% (*w*/*w*) salt. After being crushed and mixed, the raw materials were placed in a clean fermentation tank. The mouth of the tank was sealed, and the tank was placed upside down in a water basin. The chilies were naturally fermented for 90 days in Tong zi, Guizhou Province, China. Eight samples were collected on days 0, 7, 15, 22, 30, 45, 60, and 90 of fermentation and denoted as D0, D7, D15, D22, D30, D45, D60, and D90, respectively. The samples were kept at 4 °C for the measurement of physicochemical indices, FAAs, organic acids, and volatile compounds. For DNA extraction and Illumina MiSeq sequencing, the samples (20 g) were promptly kept at 80 °C after being collected at different fermentation cycles. 

### 2.3. Physicochemical Properties 

The moisture content in Zhalajiao was measured through direct drying [7]. The pH was determined with a pH meter (PHS-3C, Shanghai Instrument and Electronics Science Instrument Co., Ltd., Shanghai, China) in reference to Xu et al. [8]. Total acid was determined in accordance with [9]. Reducing sugars was measured by 3,5-dinitrosalicylic acid colorimetry [10]. Total soluble sugars were measured using anthrone colorimetry in accordance with [11].

### 2.4. Determination of Organic Acids and FAAs

Organic acids were examined using high-performance liquid chromatography. The chromatographic conditions were as described by Xiao et al. [12] with slight modifications. Separations were performed on Agilent AAAC18 (4.6 mm × 150 mm i.d., 5 μm). The column temperature was set at 28 °C. The mobile phase comprised 0.01 mol/L diammonium hydrogen phosphate (pH 2.5) and methanol (98:2, *v*:*v*), and the flow rate was 0.6 mL/min. The injection volume was 10 μL, and the detection wavelength was 220 nm.

The FAAs in Zhalajiao were detected with an automatic amino acid analyzer in accordance with a previous report with appropriate modifications [13]. The sample of Zhalajiao was settled with 2.0 mL of 10% sulfosalicylic acid for 2 h at 4 °C and then centrifuged at 8000 r/min for 1 min. Then, 1.0 mL of the supernatant fluid was filtrated using a 0.22 μm microporous membrane and determined on the L-8900 automatic amino acid analyzer (Hitachi, Tokyo, Japan). Various FAA standards were used for quantification.

### 2.5. Analysis of Volatile Compounds 

#### 2.5.1. Determination of Volatiles

Volatile compounds were determined in reference to the method of Chen et al. [14] with slight modifications. In total, 5 g of samples was placed in a 20 mL headspace injection vial containing 10 μL of 2-octanol standard with a mass concentration of 500 mg/L as an internal standard. Then, the vial was equilibrated for 30 min at 50 °C in a thermostat water bath. Next, an HS-SPME fiber (100 μm PDMS) was inserted into the vial to absorb the volatile compounds for 40 min. Finally, the head of the fiber was desorbed for 5 min at 250 °C in a gas chromatographic mass spectrometer (GCMS-GP2010, Shimadzu, Kyoto, Japan). The chromatographic column used was a DB-5MS capillary column with a size of 30 m × 0.25 mm × 0.25 μm, and the column carrier gas was helium at a flow rate of 1.0 mL/min. The temperature program was 40 °C/2 min–(10 °C/min)–150 °C/2 min–(4 °C/min) –250 °C/5 min. The mass spectrum was handled in the electron ionization mode at 70 eV with an ion-source temperature of 250 °C and a scan range from 40 *m*/*z* to 400 *m*/*z*. Volatile compounds were qualitatively analyzed using a combination of spectral library searches such as NIST05 and NIST08 (compounds with matches > 80) and retention indices. The retention index (RI) of each volatile component was calculated from the retention time of C7~C40 n-alkanes [13]. The quantification of each volatile component in the samples was performed using the internal standard method. The calculation formula was as follows:(1)$$Ci=\frac{\rho \times V\times Ai}{A\times m}$$
where *Ci* is the concentration of the identified compound (μg/kg), *ρ* is the mass concentration of the internal standard (μg/μL); *V* is the volume of the internal standard (μL); *Ai* is the peak area of each volatile component; *A* is the peak area of the internal standard substance; and *m* is the mass of the sample (kg) [8]. 

#### 2.5.2. Calculation of Odor Activity Value

Odor activity value (OAV) was applied to characterize the contribution of various aroma compounds to the main aroma components in Zhalajiao. It was determined by dividing the concentration of each compound in water by its detection threshold [4]. OAV > 1 indicated that the compound is conductive to the aroma of the sample and is a key volatile component. A high OAV is indicative of the high individual contribution of a compound [13].

### 2.6. Microbiological Analysis 

#### 2.6.1. DNA Extraction and PCR Amplification

Total DNA was extracted from each Zhalajiao sample by using an E.Z.N.A Soil DNA Kit (Omega Bio-Tek, Norcross, GA, USA) in accordance with the specification. The concentration and pureness of extracted DNA were tested using Nanodrop 2000 (Thermo Fisher Scientific, Waltham, MA, USA) and 1% agarose gels. In reference to a previous report with appropriate modifications [8], the V3–V4 variable regions of the bacterial 16S rDNA genes were amplified using primers 338F and 806R following the parameters: pre-denaturation at 95 °C for 3 min; 27 cycles at 95 °C for 30 s, 55 °C for 30 s, and 72 °C for 45 s; and final extension at 72 °C for 10 min. Each amplification was supplemented with ddH2O to a volume of 20 μL and included 4 µL of 5× FastPfu Buffer, 2 µL of 2.5 mM dNTPs, 0.8 µL of primers (5 μM), 4 µL of FastPlu Polymerase, and 10 ng of template DNA. The fungal ITS1–ITS2 gene regions were amplified with ITS1F and ITS2R primers in accordance with Chen et al. [14]. The PCR was implemented as follows: pre-denaturation at 95 °C for 3 min; 35 cycles at 95 °C for 30 s, 55 °C for 30 s, and 72 °C for 45 s; and final extension at 72 °C for 10 min. Each amplification was supplemented with ddH2O to a volume of 20 μL and included 2 µL of 10× buffer, 2 μL of 2.5 mM dNTPs, 0.8 µL of primers (5 μM), 0.2 µL of rTaq polymerase, 0.2 μL of BSA, and 10 ng of template DNA.

#### 2.6.2. Sequencing and Data Analysis

The amplified PCR products were examined by electrophoresis with 2% agarose gels and purified using an AxyPrep DNA Gel Extraction Kit (Axygen Biosciences, Union City, CA, USA). After purification and building of the paired-end library, sequencing was performed using an Illumina HiSeq 2500 platform (Majorbio Bio-pharm Technology Co., Ltd., Shanghai, China). The sequence data were analyzed and quality filtered using QIIME (version 1.17). After being merged by FLASH and quality filtered, the sequences were clustered into operational taxonomic units (OTUs) at a 97% similarity threshold by UPARSE, and UCHIME was employed to identify chimeras in accordance with Wang et al. [15]. 

### 2.7. Statistical Analysis 

All the experiments were repeated three times in parallel, and values were expressed as mean ± sd. SPSS statistical software 20.0 (IBM, Chicago, IL, USA) was applied for statistical analysis, and Pearson correlation coefficients between microorganisms and flavor substances were calculated. Significant differences among groups were determined by a one-way analysis of variance with Tukey’s honest significant difference post hoc test (*p* < 0.05). Figures were generated with Origin 2021 8.6 software. The correlation clustering heat map and networks were performed using the Lianchuan Biological Cloud Platform (https://www.omicstudio.cn/tool accessed on 10 May 2023).

## 3. Results and Discussion

### 3.1. Changes in Physicochemical Properties and Organic Acids during Fermentation 

The physicochemical indices and organic acid determination results of all samples (D0, D7, D15, D22, D30, D45, D60, and D90) during Zhalajiao fermentation are shown in Table 1. The results showed that during fermentation, the moisture content of Zhalajiao increased from 45.03% (D0) to 51.31% (D90). The pH and total acid content of the samples also changed significantly (*p* < 0.05). The raw chili had an original pH of 4.66 ± 0.01, which decreased to an ultimate pH of 4.21 ± 0.03 after 90 days of fermentation. The total acid content showed the lowest value on day 0 (4.32 ± 0.06 g/kg), increased conspicuously to its highest value on day 45 (10.26 ± 0.06 g/kg), then dropped to 9.29 ± 0.03 g/kg on day 90. As critical indicators of fermented vegetables, pH and total acids endow Zhalajiao with a distinctive sour taste that reflects and is influenced by the growth and metabolism of microorganisms during fermentation [16]. Furthermore, the growth of harmful microorganisms can be inhibited and shelf life can be extended by using the appropriate pH and acidity levels [17]. Total soluble and reducing sugars not only impart sweetness to foods but are also important nutrients for the survival of microorganisms [18]. In this study, during the first 15 days of fermentation, a large number of microorganisms utilized sugar fermentation to produce acid, resulting in a sharp decrement in the content of reducing acids from 1.25 ± 0.02 g/kg (D0) to 0.57 ± 0.02 g/kg (D15). However, the reducing sugar content did not change significantly after 45 days of fermentation, likely due to the inhibition of microbial activity with the increase in acidity. Furthermore, during days 0–15 of fermentation, the total soluble sugar content in Zhalajiao did not change significantly and remained at approximately 3.46 g/kg because microorganisms mainly use reducing sugars as carbon and energy sources. In the later stage of fermentation, the content of reducing sugars in Zhalajiao cannot meet the growth and metabolism requirements of microorganisms, leading to the decomposition and utilization of polysaccharides by microorganisms. The total soluble sugar content decreased from 3.54 ± 0.10 g/kg (D22) to 3.20 ± 0.05 g/kg (D45) and remained stable thereafter.

Acidity is a very important taste quality of Zhalajiao. Although Zhalajiao fermentation produces a variety of acidic substances, organic acids, such as lactic, acetic, malic, and succinic acids, are the main contributors to acidity. The proportion of organic acids also helps improve the flavor and balance of fermented chili paste [13]. The total organic acids in Zhalajiao first increased and then decreased during fermentation and reached the maximum value of 13.92 g/kg on day 15 of fermentation (Table 1). The contents of oxalic and malic acids decreased continuously during the first 22 days of fermentation, increased and stabilized after 30 days of fermentation, and significantly decreased after 60 days of fermentation (*p* < 0.05). The content of tartaric acid increased continuously during fermentation and remained at approximately 1.1 g/kg after 30 days of fermentation. Succinic acid had an initial content of 0.78 ± 0.041 g/kg and peaked at 3.60 ± 0.064 g/kg on day 15 of fermentation but could not be detected after day 30. The lactic acid content reached a maximum of 3.85 ± 0.073 g/kg on day 15 of fermentation and then declined slightly to 3.21 ± 0.024 g/kg. On day 90 of Zhalajiao fermentation, lactic acid accounted for 33.72% of the total organic acid content, whereas acetic acid accounted for 29.62%. Consistent with our results, the findings of Xu et al. [8] showed that lactic and acetic acids are the most important organic acids in chili paste. Lactic acid is an important metabolite of microbial cells, and lactic acid produced by microbial fermentation is used as a souring agent and preservative in food because of its mellow taste. Two types of lactic acid fermentation exist: homolactic fermentation and heterolactic fermentation. In homolactic fermentation, LAB uses glucose to produce lactic acid through glycolysis. In heterolactic fermentation, ethanol or acetic acid are produced in addition to lactic acid [19]. Lactic acid accounted for the largest proportion of organic acids in Zhalajiao. As Zhalajiao fermentation proceeded, the lactic acid content continued to increase, whereas the acetic acid content decreased, indicating that Zhalajiao fermentation is mainly homolactic fermentation. 

### 3.2. Changes in FAAs during Fermentation

FAAs not only have nutritional functions, but they are also important flavoring substances. Asp and Glu are umami amino acids; His, Arg, Val, Phe, Leu, Met, Ile, and Tyr are slightly bitter amino acids; and Ser, Thr, Gly, and Ala are classified as sweet amino acids. These FAAs are composed of different taste-related peptides, and changes in amino acid composition may have a certain effect on aromatic components, such as alcohols and esters [13]. Table 2 shows that 17 FAAs were detected during Zhalajiao fermentation. The FAA content had an initial value of 10.88 g/kg, which peaked at 16.86 g/kg on day 7, and then decreased. Similar results were reported for fermented fish-chili paste [20] and may be attributed to lactic acid fermentation, in which soluble proteins were degraded into FAAs by endogenous enzymes secreted by microorganisms in Zhalajiao during fermentation [21]. The inhibition of the growth and metabolism of microorganisms with the progress of fermentation reduced the decomposition rate of proteins. At the same time, FAAs participated in the reaction for the formation of flavor substances, resulting in a reduction in the FAA content from 16.86 g/kg (D7) to 3.74 g/kg (D45). Subsequently, the FAAs content remained relatively stable at approximately 4 g/kg from days 45 to 90 of fermentation. At the beginning of fermentation, sweet amino acids dominated among the amino acids, contributing to the flavor of fresh chili. As fermentation progressed, sweet amino acids were consumed and decreased, whereas bitter, umami, and tasteless amino acids increased. Ser is the main sweet amino acid. It increased first and then decreased during fermentation. Ser was present at the highest content in the first 30 days of fermentation, accounting for 70.42–86.18% of the total content of amino acids. Different from our study, a previous work showed that Thr, followed by Asp and Ser, was the most abundant amino acid in Paojiao [13]. The difference in the amino acid contents may be ascribed to the different chili varieties and fermentation environments used. After 45 days of fermentation, the content of Ser was only approximately 9%, and the content of various FAAs was relatively balanced. Additionally, the content of His in Zhalajiao before fermentation was 43.93 ± 0.02 mg/kg and decreased to almost undetectable levels after fermentation. Given that FAAs are important components in the growth of LAB, the decrease in the His content during the late fermentation period may be due to utilization by LAB [14].

### 3.3. Changes in Volatile Flavor Compounds during Fermentation

In this study, the volatile compounds were analyzed, and the results are shown in Appendix A. Volatile compounds play an important role as flavor determinants in the quality evaluation of fermented chili foods, which are affected by microbial communities and metabolic reactions during fermentation [22]. Approximately 66 different volatile components were detected throughout the whole fermentation process of Zhalajiao. These components included 28 esters, 18 terpenes, 12 alkanes, 5 acids, 1 alcohol, 1 aldehyde, and 1 pyrazine. The content of total volatile components decreased from an original value of 36.11 mg/kg to 13.30 mg/kg on day 7 and then increased continuously, reaching the maximum value of 82.04 mg/kg on day 60. Furthermore, the analysis of the OAV of the above volatiles showed that a total of 21 aroma substances, including 13 esters, 7 terpenes, and 1 pyrazine, had an OAV > 1 during Zhalajiao fermentation (Table 3). Although these substances did not have the highest content in the whole flavor system, they had a significant effect on the flavor quality of Zhalajiao [20]. Ethyl caproate, (Z)-4-decenoic acid, ethyl ester, (+)-Limonene, Ocimene, Linalool, and β-Caryophyllene had an OAV > 1 on day 0, indicating that these six substances contributed to the aroma of fresh chili. Among these substances, (+)-Limonene had an OAV > 1000, which is evidence showing that it is the major aroma component of fresh chili. Zhalajiao contained 21 compounds with an OAV > 1 after 45 days of fermentation. Of these compounds, Ethyl butyrate, Ethyl caproate, Hexyl acetate, Hexanoic acid, hexyl ester, Linalool, β-Ionone, and 2-isobutyl-3-methoxypyrazine had an OAV > 100, indicating that these seven substances were the main aroma components in Zhalajiao after fermentation. Β-Ionone was not detected in the first 22 days of fermentation. However, its content reached 965.86 μg/kg on day 45 of fermentation, likely due to the degradation of carotene during Zhalajiao fermentation, resulting in β-Ionone formation [23]. Furthermore, given its extremely low threshold, β-Ionone contributed greatly to the aroma of Zhalajiao after fermentation. Remarkably, although alkane compounds were present at high amounts in Zhalajiao, they were not the main aroma components of Zhalajiao due to their high flavor threshold and small contribution to the flavor of Zhalajiao [22].

The dynamic changes in volatile compounds in samples at different fermentation times were further explored by cluster heat mapping and principal component analysis (PCA). Cluster analysis (Figure 1A) showed that the volatile flavor compounds in Zhalajiao fermentation mainly accumulated during the 30–60-day fermentation period. During Zhalajiao fermentation, the content of esters, alcohols, aldehydes, acids, and pyrazines increased first then decreased. Esters are typically produced through the esterification of short-chain acids and alcohols and are considered to be an important source of pleasant sweet and fruity flavors in fermented foods [10]. In this study, only eight types of ester compounds were detected with a content of 5.22 mg/kg on day 0. With the progress of fermentation, the types and contents of ester compounds increased continuously, reaching the maximum content of 42.89 mg/kg on day 60 of fermentation. Terpenoids are typically present in various plant cells in the form of glycosides and are one of the secondary metabolites of plants [24]. Seven terpenoids were detected in Zhalajiao with a total content of 24.88 mg/kg at day 0 of fermentation. Although other terpenoids were detected after 30 days of fermentation, the total content of terpenoids at this time point was lower than that on day 0. Alcohols, aldehydes, acids, and pyrazines were detected after 30 days of fermentation at relatively low contents. Notably, the composition of volatile components in Zhalajiao differed at different fermentation times. Terpenoids accounted for 68.9% and esters and alkanes accounted for approximately 15% of the total content of volatile components on day 0 of fermentation. Among terpenoids, (+)-limonene was present at the highest content, accounting for approximately 40% of the total content. With the extension of fermentation time, the proportion of terpenoids decreased, whereas that of esters and alkanes increased. This change trend has also been reported previously [16]. After 30 days of fermentation, ester compounds accounted for approximately 50% and terpenoids and alkanes accounted for approximately 20% of the total content, likely because during fermentation, alcohols and acids can be used as precursors for the synthesis of ester compounds and esterification yields esters [15]. Nevertheless, fermented Zha-chili is acidic, and esters may be hydrolyzed to acids and alcohols under acidic conditions. In other words, the esterification is reversible. Such reversibility may account for the decrement in the ester content at the later stage of fermentation.

PCA was performed on the basis of the average content of all volatile compounds to visualize the dynamics of volatile compounds during fermentation. The PCA revealed that the variance contribution rate of PC1 (70.6%) and PC2 (11.9%) was 82.5%, which could well represent the overall sample information (Figure 1B). Each group had excellent repeatability, and significant discrepancies were found in the aroma characteristics of the samples, indicating that the increase in the fermentation time could dramatically affect the overall aroma characteristics of Zhalajiao during natural fermentation [25]. The PCA divided the Zhalajiao samples taken at different fermentation stages into three independent groups, among which the D0, D7, D15, D22, D30, and D90 samples showed high similarity and clustered into the same category. These samples were distant from the D45 and D60 samples in terms of distribution area, suggesting significant differences in the volatile compound content between these samples and the Zhalajiao samples collected on days 45 and 60 [26]. Additionally, the 21 key volatile compounds with an OAV > 1 in Zhalajiao fermentation were subjected to cluster analysis, as shown in Figure 1C. More types and higher contents of key volatiles, particularly esters and terpenes, were found on day 45 of fermentation than at previous stages (red areas of clusters in Figure 1C). Consequently, this stage can be deduced to be the key stage of flavor formation. Interestingly, (+)-limonene, ocimene, and β-caryophyllene have fruity, fragrant, and spicy properties, respectively [9], and are the major aroma compounds in erijingtiao chili before fermentation. Ethyl butyrate, ethyl caproate, hexyl acetate, hexanoic acid, hexyl ester, linalool, and β-ionone had significant contributions to the flavor of Zhalajiao on day 45 of fermentation, endowing fruity and grassy aromas [22]. Furthermore, the aroma of 2-isobutyl-3-methoxypyrazine, which appeared after fermentation, is similar to that of pepper and coffee [3]. Although 2-osobutyl-3-methoxypyrazine was present at low contents in Zhalajiao, it had a threshold of only 0.016 μg/kg and an OAV > 1000 and was therefore the characteristic aroma component of Zhalajiao after 30 days of fermentation. 

### 3.4. Alpha Diversity of Microorganisms

This study analyzed the alpha diversity of the microbial communities in Zhalajiao at different fermentation stages by using Illumina Miseq high-throughput sequencing technology to gain a good understanding of the characteristics and dynamic succession of microbial communities throughout Zhalajiao fermentation. A total of 940,098 valid bacterial sequences were obtained after sequence screening. The optimized sequences had an average length of 442 bp. Moreover, 892,596 valid fungal sequences, with an average length of 274 bp for the optimized sequences, were obtained. The microbial diversity in Zhajiao was analyzed at the level of 3% dissimilarity (Figure 2A).

The Ace and Chao indices represent bacterial abundance. They are used to estimate the number of species in a sample and are positively correlated with microbial abundance [27]. As shown in Figure 2A, during Zhalajiao fermentation, the Ace and Chao indices of bacteria were considerably higher than those of fungi, indicating that bacterial flora were far more abundant than fungal flora. The bacterial Ace and Chao indices were high at the beginning of Zhalajiao fermentation, and the abundance of bacterial flora fluctuated during fermentation. The fungal Ace and Chao indices first decreased, then increased, and continued to decrease throughout the entire fermentation process. Overall, the abundance of fungal communities in Zhalajiao decreased over 90 days of fermentation. The Shannon and Simpson indices reflect community diversity, with the Shannon index being positively related and the Simpson index being negatively related to the complexity of microbial flora [28]. During the fermentation of Zhalajiao, the variation range of the bacterial Shannon index was significantly greater than that of the fungal Shannon index. At the initiation of fermentation (D0), the difference between the Shannon indices of bacteria and fungi was small. However, as fermentation progressed, the bacterial Shannon index tended to increase and then decreased and remained stable after 22 days, whereas the fungal Shannon index decreased significantly and then remained stable. The trend of the Simpson index was opposite to that of the Shannon index. These data indicated that large numbers of bacteria and fungi were present at the initial stage of Zhalajiao fermentation. Nevertheless, as a result of the development of dominant flora, such as *Lactobacillus*, *Cyanobacteria*, and *Candida* during fermentation, some species decreased, resulting in a reduction in microbial diversity. In general, bacterial communities showed greater diversity and richness than fungal communities. 

### 3.5. Changes in Bacterial Community Diversity and Composition during Fermentation

A total of 60 different bacterial genera were identified during Zhalajiao fermentation. The taxonomic group with a relative abundance of more than 1% was defined as the dominant group [3] and included six species, namely, *Lactobacillus*, *Cyanobacteria*, *Staphylococcus*, *Mitochondria*, *Enterobacter*, and *Xanthomonas*. The relative abundance of bacterial communities at the genus level in Zhalajiao at different fermentation times is shown in Figure 2B. In the beginning, the abundance of Lactobacillus in Zhalajiao was only 1.99%, whereas the abundances of *Cyanobacteria*, *Staphylococcus*, and *Mitochondria* were 46.11%, 45.82%, and 1.95%, respectively. After 7 days of fermentation, the abundance of *Lactobacillus* increased rapidly to 64.23%, that of *Cyanobacteria* decreased to 29.65%, and that of *Staphylococcus* was only 2.21%. Subsequently, the abundance of *Lactobacillus* ranged from 64% to 72%, whereas that of *Cyanobacteria* ranged from 16% to 33%. As a result of the increase in oxygen consumption and acid production, *Lactobacillus* occupied a dominant position throughout the entire fermentation process. This result was consistent with the finding reported by Dong et al. [4]. Notably, *Lactobacillus* is the dominant microorganism in many fermented foods, including Koumiss [29], Chinese fermented vegetables [5], and vinegar [30]. *Staphylococcus* has been found to be the predominant genus in fermented foods, including Chinese strong-flavored liquor [31] and traditional Chinese low-salt fermented fish [32]. It has been demonstrated that *Staphylococci* from raw milk and cheese exhibits lipolytic and proteolytic activities and creates significant amounts of flavor compounds, such as FAAs, aldehydes, amines, and free fatty acids [33].

### 3.6. Changes in Fungal Community Diversity and Composition during Fermentation

A total of 61 different fungal genera were identified during Zhalajiao fermentation. Five species had a relative abundance greater than 1%: *Candida*, *Meyerozyma*, *Aspergillus*, *Gibberella*, and *Fusarium*. The relative abundance of fungi at the genus level in Zhalajiao at different fermentation times is shown in Figure 2C. On day 0 of fermentation, the abundance of *Candida* was 87.46%, that of *Meyerozyma* was 6.9%, and that of *Gibberella* and *Fusarium* was 1%. After 7 days of fermentation, the abundance of *Candida* remained above 98%. *Candida* was the dominant fungus in the fermentation of Zhalajiao. *Candida* and *Meyerozyma* have been reported to be critical for the flavor formation of fermented chili paste [8] and pickled radishes [34]. In addition, *Meyerozyma* is one of the main yeasts found in fermented soybean paste and koji from Japan and China [35]. It can produce volatile phenolic compounds during the fermentation of Thai soy sauce and improve the flavor characteristics of Thai soy sauce [36].

### 3.7. Pearson Correlation Analysis between Core Microorganisms and Flavor Components during Fermentation 

In this study, 17 amino acids, 6 organic acids, and 21 key volatile compounds with a crucial role in the taste and flavor of Zhalajiao were detected during fermentation. Remarkably, given that fermentation is a process based on microbial metabolism, microorganisms are closely related to the production of typical fermented flavor substances [8]. The potential relationship between microbial community succession and significant changes in key flavor components during Zhalajiao fermentation was revealed by using Pearson correlation analysis (|r| > 0.70, *p* < 0.05).

Figure 3A,B show the correlation between the succession of dominant bacterial communities and the changes in key flavor compounds during Zhalajiao fermentation. The diagram shows that *Lactobacillus* was mainly positively correlated with most organic acids and FAAs but was weakly correlated with volatile compounds. Interestingly, *Lactobacillus* was significantly associated with lactic acid (*p* < 0.05). Lactic acid provides sourness and can interact with alcohols, aldehydes, and ketones during the fermentation of many vegetables to produce a variety of new flavor substances [16]. The results of this study are consistent with those of Fang et al. [30], who reported that *Lactobacillus* was significantly positively correlated with the various flavor compounds (including lactic acid, acetic acid, ethyl ester, dodecanoic acid, and 2,4-di-tert-butylphenol) produced during the fermentation of Zhejiang rose vinegar. Furthermore, *Lactobacillus* has been reported to produce D-amino acids, which can enhance the taste of fermented foods [37]. *Enterobacter* exhibited a significant positive correlation with some esters with high odor activity. By contrast, *Staphylococcus* was closely correlated with histidine (bitterness) (*p* < 0.01) likely because it can secrete bitter amino acids and peptides [38]. In addition, *Staphylococcus* showed a significantly positive correlation with (+)-limonene and ocimene. *Xanthomonas* presented a significantly positive correlation with succinic acid. However, different from our study, the work of Xu et al. [8] found that *Xanthomonas* was positively correlated with phenol and styrene. These results indicated that *Lactobacillus*, *Enterobacter*, and *Staphylococcus* are the core bacteria that may play an important role in the production of flavor compounds. Rao et al. [34] also proposed that *Lactobacillus* and *Enterobacter* contributed significantly to the flavor of pickled radish. 

As shown in Figure 3C,D, a significant correlation (*p* < 0.05) was found between 4 fungi and 17 flavor substances in Zhalajiao fermentation. The network graph shows that fungi mainly exhibited significant correlations with changes in amino and organic acids during Zhalajiao fermentation. In the fungal community, Candida was conspicuously correlated with 14 flavor substances in fermentation, suggesting that it is the main fungal genus responsible for the formation of Zhalajiao flavor. Specifically, *Candida* showed a high correlation with various amino acids, such as Val, Phe, Leu, Lys, Ile, Tyr, Thr, and Gly (*p* < 0.05). This result has also been reported by Xiao et al. [12], who pointed out that *Candida* plays an important role in the production of amino acids (Asp, Gly, Leu, Ile, His, Ser, and Glu) in fermented foods. *Candida* etchellsii have been reported to enhance the flavor of soy sauce [39]. Furthermore, our results demonstrated that *Meyerozyma*, *Gibberella*, and *Fusarium* were highly positively correlated with His (bitter FAAs) and (+)-limonene (with citrus and lemon aroma) (*p* < 0.01), positively correlated with Arg, acetic acid, and ocimene, but were negatively correlated with other FAAs and organic acids. These results indicate that *Candida* is mainly the core fungus that produces the key flavor of Zhalajiao.

## 4. Conclusions

The correlation between the flavor components and microbial composition during Zhalajiao fermented for 90 days was elucidated. During Zhalajiao fermentation, pH decreased, the total acid content continuously increased, and the organic acid content first increased then decreased. Lactic and acetic acids were the main organic acids. The content of total FAAs decreased after fermentation. Bitter, umami, and tasteless FAAs increased, whereas sweet FAAs decreased after fermentation. In addition, 21 volatile compounds (OAV > 1), mainly esters and terpenoids, were identified as the key volatile compounds of Zhalajiao. Finally, high-throughput sequencing and correlation analysis revealed that in Zhalajiao, *Lactobacillus* and *Candida* were the most dominant bacterial and fungal genera, respectively. These genera were significantly positively correlated with most of the key flavor components and had important contributions to the flavor formation of Zhalajiao during natural fermentation. These results contribute to improving the quality of Zhalajiao and provide theoretical guidance for stabilizing the flavor of Zhalajiao.

## Figures and Tables

**Figure 1 foods-12-03849-f001:**
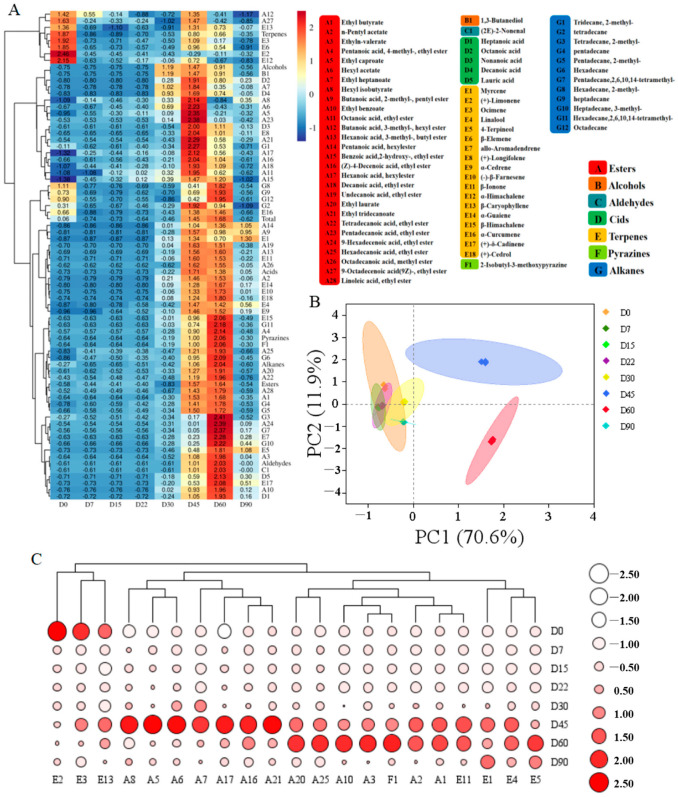
Dynamics in volatile compounds of Zhalajiao at different fermentation stages. Heatmap of the volatile compounds (**A**), principal component analysis (**B**), heatmap cluster analysis of the 21 key volatile compounds (**C**). The heatmap shows the relative abundance of volatiles at different stages of fermentation, which is directly correlated with the color intensity.

**Figure 2 foods-12-03849-f002:**
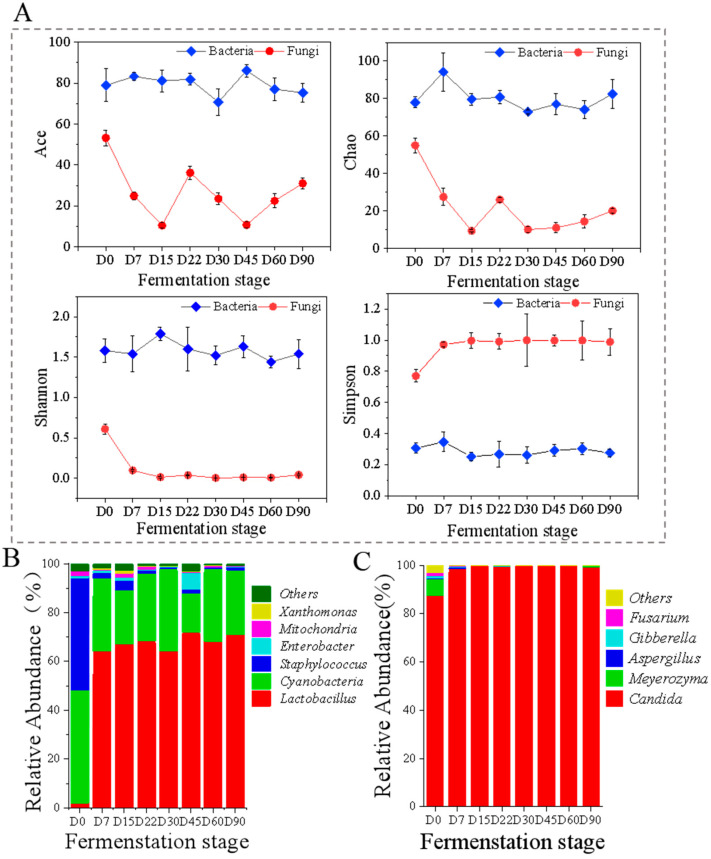
Dynamics in microbial community of Zhalajiao at different fermentation stages. Alpha diversity (**A**); the relative abundance of bacterial community (**B**) and fungal community (**C**) at genus level. Some classes with relative abundance < 1% are regarded as “others”.

**Figure 3 foods-12-03849-f003:**
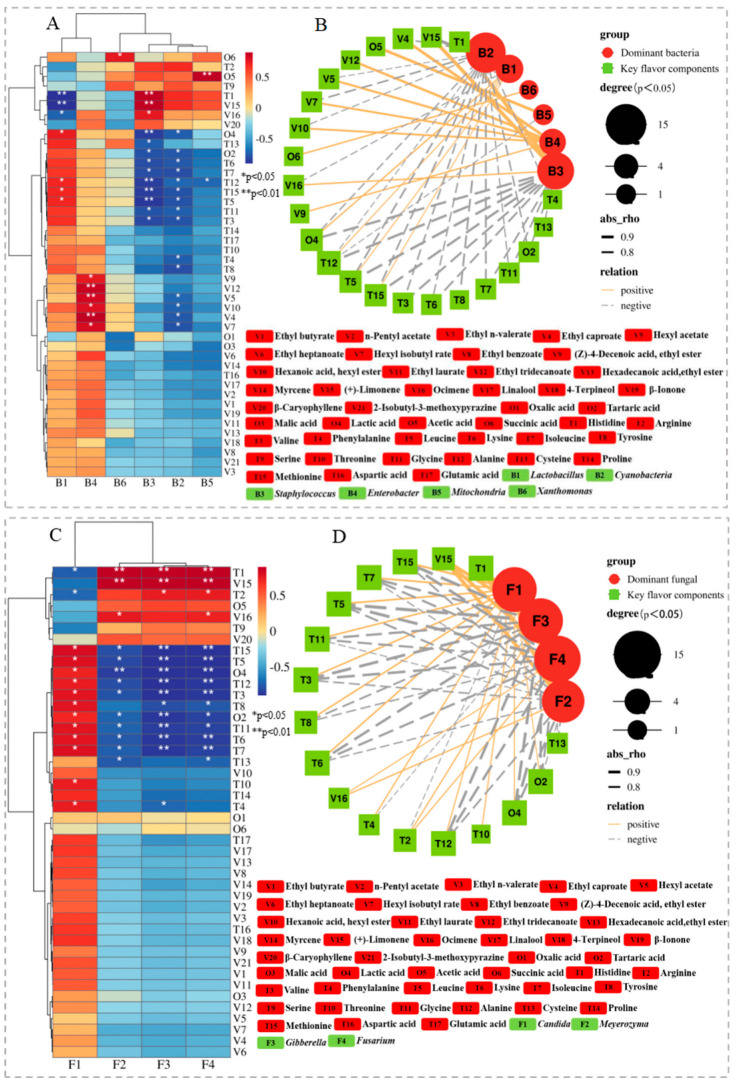
The clustering heatmap (bacteria: (**A**); fungi: (**C**)) and correlation network (bacteria: (**B**); fungi: (**D**)) between microbial community at genus level and main flavor compounds. The circular and square symbols indicate the preponderant genus and the key flavor components, respectively. Orange and grey lines show a positive and negative correlation, respectively.

**Table 1 foods-12-03849-t001:** Changes in physiochemical properties during the fermentation of Zhalajiao.

Sample		Content							
		D0	D7	D15	D22	D30	D45	D60	D90
Physiochemical properties	Moisture content (%)	45.03 ± 0.30 ^d^	45.48 ± 0.36 ^d^	45.85 ± 0.79 ^cd^	45.78 ± 0.34 ^cd^	45.52 ± 0.90 ^d^	47.38 ± 1.71 ^bc^	48.24 ± 1.44 ^b^	51.31 ± 0.97 ^a^
	pH	4.66 ± 0.01 ^a^	4.16 ± 0.01 ^c^	4.14 ± 0.01 ^c^	4.17 ± 0.01 ^c^	4.05 ± 0.01 ^e^	4.09 ± 0.01 ^d^	4.08 ± 0.01 ^d^	4.21 ± 0.03 ^b^
	Total acids (g/kg)	4.32 ± 0.06 ^g^	5.48 ± 0.03 ^f^	5.90 ± 0.13 ^e^	6.95 ± 0.03 ^d^	9.61 ± 0.10 ^b^	10.26 ± 0.06 ^a^	9.70 ± 0.10 ^b^	9.29 ± 0.03 ^c^
	Reducing sugars (g/kg)	1.25 ± 0.02 ^a^	1.23 ± 0.01 ^a^	0.57 ± 0.02 ^b^	0.56 ± 0.02 ^bc^	0.52 ± 0.02 ^c^	0.42 ± 0.02 ^d^	0.42 ± 0.02 ^d^	0.41 ± 0.02 ^d^
	Total soluble sugars (g/kg)	3.46 ± 0.03 ^a^	3.46 ± 0.02 ^a^	3.45 ± 0.04 ^a^	3.54 ± 0.10 ^a^	3.49 ± 0.09 ^a^	3.20 ± 0.05 ^b^	3.18 ± 0.08 ^b^	3.20 ± 0.06 ^b^
Organic acids (g/kg)	Oxalic acid	0.61 ± 0.002 ^d^	0.43 ± 0.001 ^e^	0.40 ± 0.002 ^f^	0.38 ± 0.001 ^f^	0.64 ± 0.60 ^c^	0.70 ± 0.013 ^b^	0.74 ± 0.001 ^a^	0.64 ± 0.017 ^c^
	Tartaric acid	0.49 ± 0.009 ^e^	0.72 ± 0.007 ^d^	0.87 ± 0.010 ^c^	0.86 ± 0.007 ^c^	1.12 ± 0.011 ^ab^	1.14 ± 0.019 ^a^	1.12 ± 0.010 ^ab^	1.10 ± 0.464 ^b^
	Malic acid	1.52 ± 0.009 ^c^	1.39 ± 0.023 ^d^	1.29 ± 0.068 ^f^	1.34 ± 0.036 ^e^	1.81 ± 0.050 ^b^	1.82 ± 0.053 ^b^	1.90 ± 0.017 ^a^	1.84 ± 0.092 ^b^
	Lactic acid	1.40 ± 0.071 ^h^	2.41 ± 0.023 ^g^	3.85 ± 0.073 ^a^	3.72 ± 0.001 ^b^	3.65 ± 0.166 ^c^	3.48 ± 0.127 ^d^	3.42 ± 0.006 ^e^	3.21 ± 0.024 ^f^
	Acetic acid	4.02 ± 0.056 ^a^	3.34 ± 0.025 ^d^	3.91 ± 0.004 ^b^	3.83 ± 0.046 ^c^	2.73 ± 0.040 ^h^	3.00 ± 0.040 ^e^	2.95 ± 0.002 ^f^	2.82 ± 0.006 ^g^
	Succinic acid	0.78 ± 0.041 ^d^	3.18 ± 0.012 ^c^	3.60 ± 0.064 ^a^	3.41 ± 0.021 ^b^	nd ^e^	nd ^e^	nd ^e^	nd ^e^
	Total	8.82	11.47	13.92	13.54	9.95	10.14	10.13	9.52

Note: Different letters in the same row represent significant differences (*p* < 0.05). nd: not detected.

**Table 2 foods-12-03849-t002:** Changes in free amino acids (FAAs, mg/kg) during the fermentation of Zhalajiao.

FAAs (mg/kg)	D0	D7	D15	D22	D30	D45	D60	D90
Bitter AA	797.24	1133.59	1231.81	1083.93	1327.66	1480.18	1510.28	1619.92
Histidine	43.93 ± 0.02 ^a^	0.10 ± 0.07 ^c^	nd ^d^	nd ^d^	0.44 ± 0.01 ^b^	nd ^d^	nd ^d^	nd ^d^
Arginine	207.09 ± 1.68 ^b^	215.31 ± 4.88 ^a^	2.55 ± 0.23 ^e^	2.06 ± 0.48 ^e^	24.14 ± 1.84 ^d^	37.19 ± 1.65 ^c^	37.46 ± 2.00 ^c^	26.67 ± 1.77 ^d^
Valine	111.53 ± 0.47 ^g^	195.69 ± 7.92 ^f^	254.83 ± 3.58 ^d^	222.01 ± 1.58 ^e^	269.24 ± 9.97 ^c^	269.60 ± 5.73 ^c^	283.23 ± 3.87 ^b^	322.97 ± 1.10 ^a^
Phenylalanine	140.89 ± 1.18 ^f^	161.36 ± 5.80 ^e^	227.93 ± 5.53 ^c^	196.28 ± 1.14 ^d^	231.34 ± 5.07 ^c^	300.05 ± 6.85 ^b^	296.80 ± 0.65 ^b^	317.82 ± 2.32 ^a^
Leucine	91.35 ± 5.76 ^g^	238.35 ± 2.95 ^f^	325.88 ± 0.84 ^d^	286.01 ± 3.14 ^e^	353.63 ± 3.41 ^c^	355.85 ± 1.95 ^c^	372.45 ± 14.51 ^b^	414.59 ± 2.66 ^a^
Methionine	31.46 ± 0.17 ^g^	68.25 ± 0.59 ^f^	89.17 ± 0.75 ^d^	82.46 ± 3.10 ^e^	97.17 ± 2.08 ^c^	104.03 ± 7.26 ^b^	108.05 ± 6.49 ^b^	116.13 ± 3.08 ^a^
Isoleucine	53.91 ± 3.19 ^g^	113.64 ± 2.30 ^f^	145.72 ± 0.68 ^d^	127.11 ± 1.22 ^e^	156.19 ± 12.20 ^c^	184.92 ± 3.34 ^c^	184.78 ± 2.93 ^b^	203.78 ± 2.09 ^a^
Tyrosine	117.08 ± 2.73 ^g^	140.89 ± 7.02 ^f^	185.73 ± 3.84 ^d^	168.00 ± 0.89 ^e^	195.51 ± 2.50 ^c^	228.54 ± 5.62 ^a^	227.51 ± 1.49 ^a^	217.96 ± 4.55 ^b^
Sweet AA	9502.24	15,063.93	8654.07	9216.78	7348.06	1136.73	1153.39	1211.20
Serine	9255 ± 32.00 ^b^	14,530 ± 265.00 ^a^	8080 ± 128.00 ^d^	8676 ± 303.00 ^c^	6697 ± 272.00 ^e^	342 ± 19.00 ^f^	355 ± 21.00 ^f^	325 ± 40 ^f^
Threonine	39.12 ± 2.55 ^g^	77.06 ± 2.93 ^f^	94.76 ± 2.28 ^d^	88.06 ± 1.14 ^e^	105.77 ± 4.66 ^c^	231.97 ± 3.05 ^a^	227.48 ± 2.74 ^a^	182.47 ± 3.36 ^b^
Glycine	46.54 ± 4.94 ^g^	93.44 ± 3.19 ^f^	114.88 ± 3.55 ^d^	106.04 ± 4.60 ^e^	123.80 ± 2.88 ^c^	134.61 ± 4.47 ^b^	136.71 ± 3.30 ^b^	172.77 ± 2.38 ^a^
Alanine	161.58 ± 8.11 ^d^	363.43 ± 30.48 ^c^	364.43 ± 30.48 ^c^	346.68 ± 5.96 ^c^	421.49 ± 4.56 ^b^	428.15 ± 3.69 ^b^	434.20 ± 1.52 ^b^	530.96 ± 14.77 ^a^
Umami AA	291.17	284.17	319.12	324.32	365.91	633.45	637.74	686.10
Aspartic acid	52.63 ± 1.97 ^d^	38.21 ± 1.42 ^e^	24.71 ± 2.22 ^f^	76.02 ± 2.18 ^c^	57.46 ± 1.40 ^d^	224.78 ± 10.55 ^a^	220.57 ± 6.67 ^a^	204.88 ± 4.26 ^b^
Glutamic	238.54 ± 14.94 ^e^	245.96 ± 4.84 ^e^	294.41 ± 1.19 ^d^	248.3 ± 5.52 ^e^	308.45 ± 6.35 ^c^	408.67 ± 0.39 ^b^	417.17 ± 4.15 ^b^	481.22 ± 5.27 ^a^
Tasteless AA	288.98	378.47	444.47	393.33	465.34	490.62	508.60	556.99
Cysteine	0.94 ± 0.30 ^ab^	9.41 ± 1.26 ^ab^	10.08 ± 2.63 ^a^	10.76 ± 1.22 ^a^	11.47 ± 1.67 ^a^	6.33 ± 0.83 ^c^	4.77 ± 0.65 ^c^	7.68 ± 0.37 ^bc^
Proline	172.36 ± 7.79 ^f^	184.50 ± 3.66 ^e^	223.46 ± 5.29 ^c^	198.70 ± 12.35 ^d^	226.32 ± 3.01 ^c^	240.66 ± 8.43 ^b^	246.80 ± 0.97 ^b^	282.83 ± 1.06 ^a^
Lysine	115.68 ± 1.59 ^g^	184.56 ± 3.66 ^f^	210.93 ± 2.85 ^e^	183.87 ± 3.21 ^f^	227.55 ± 5.31 ^d^	243.63 ± 3.84 ^c^	257.03 ± 4.46 ^b^	266.48 ± 0.62 ^a^
Total AA (g/kg)	10.88	16.86	10.65	11.02	9.51	3.74	3.81	4.07

Note: Different letters in the same row represent significant differences (*p* < 0.05). nd: not detected.

**Table 3 foods-12-03849-t003:** The average OAV values of major volatile compounds in different fermentation stages.

Volatile Compounds	Odor Threshold (μg/kg)	Odor Active Value (OAV)							
		D0	D7	D15	D22	D30	D45	D60	D90
Ethyl butyrate	1	nd	nd	nd	nd	137.87	876.56	938.81	118.00
n-Pentyl acetate	43	nd	nd	nd	nd	16.42	36.95	37.95	11.98
Ethyl n-valerate	5	nd	nd	nd	nd	4.83	65.84	100.21	26.26
Ethyl caproate	1	15.00	189.68	292.56	377.55	461.05	1429.51	333.51	284.53
Hexyl acetate	2	nd	nd	42.80	52.98	269.67	576.77	47.77	70.35
Ethyl heptanoate	2.2	nd	nd	nd	nd	50.89	74.18	31.92	19.76
Hexyl isobutyl rate	13	nd	8.87	5.87	7.37	13.31	30.23	2.35	13.51
Ethyl benzoate	60	nd	nd	nd	nd	0.68	1.48	2.39	0.77
(Z)-4-Decenoic acid, ethyl ester	150	3.61	3.97	4.37	5.26	6.87	22.96	15.81	3.99
Hexanoic acid, hexyl ester	10	nd	48.94	40.44	53.17	57.02	157.96	86.36	40.68
Ethyl laurate	400	0.38	0.37	0.50	0.48	0.75	2.83	3.65	0.43
Ethyl tridecanoate	180	nd	nd	nd	nd	0.12	3.09	1.22	0.27
Hexadecanoic acid, ethyl ester	2000	0.02	1.19	1.25	1.27	1.01	5.62	7.60	0.51
Myrcene	15	nd	nd	nd	nd	11.88	26.26	18.61	25.82
(+)-Limonene	10	1413.31	5.52	5.91	21.54	12.30	82.05	167.83	67.31
Ocimene	34	63.54	10.03	10.52	15.32	14.74	46.87	28.20	9.06
Linalool	6	45.34	57.21	59.17	92.55	118.05	421.56	412.67	275.33
4-Terpineol	40	nd	nd	nd	nd	0.94	4.20	8.78	6.26
β-Ionone	0.007	nd	nd	nd	nd	38,934	137,980	133,601	29,174
β-Caryophyllene	64	49.45	12.09	4.76	13.18	8.23	48.41	37.95	23.43
2-Isobutyl-3-methoxypyrazine	0.016	nd	nd	nd	nd	1868.82	6755.24	11,141	1405.22

Note: nd: not detected. Odor thresholds were mainly obtained from an online database (http://www.leffingwell.com (accessed on 19 August 2023)) with water applied as the matrix.

## Data Availability

Data are contained within the article.

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
