# Peer review of "Relationship between the Dynamics of Flavor Compounds and Microbial Succession in the Natural Fermentation of Zhalajiao, a Popular Traditional Chinese Fermented Chili Paste"

_foods, 2023, doi:10.3390/foods12203849_

Round 1

Reviewer 1 Report

This paper describes the composition changes and the evolution in the microbial succession during the fermentation of Zhalajiao, a traditional Chinese chili paste. In my opinion, the manuscript is very interesting, since the Authors performed a comprehensive analysis of the flavor active compounds composition and microbial evolution, and the observed modifications were correlated and adequately discussed.Specific comments are as follows:

Please, check the microorganisms names in line 58 (Companilatobacter and Lactiplantabacter)

Line 103: please change “mensuration” to “measurement”

2.5.1 Line 134: please change “an aged extraction head” to “a HS-SPME fiber”.

Please, specify how the compounds identification was performed. Probably, a database to compare the obtained spectra was used. Did you perform analyses of standard compounds? If only one method is used, you should specify the “tentative” compounds identification.

2.7 lines 181-182: which parameter related to microorganisms? DNA amount?

Lines 335-336: please, add an explanation for esterification reversibility in Zhalajiao storage conditions.

References and description for the calculation of the indexes listed in Figure 2, in my opinion, belong to the  Material and Methods section.

Lines 441-443: how was the role of the listed compounds in the taste and flavour determined, without performing sensory analysis?

Reviewer 2 Report

Reviewer comments: Foods

Relationship between the dynamics of flavor compounds and microbial succession in the natural fermentation of Zhalajiao, a popular traditional Chinese fermented chili paste

I am pleased to provide my comments on the article titled "Relationship between the dynamics of flavor compounds and microbial succession in the natural fermentation of Zhalajiao, a popular traditional Chinese fermented chili paste" I hope that these comments will be helpful to the authors in improving their work.

General Comment:

Overall, this paper is quite intriguing. The authors aim to elucidate the formation of essential flavor compounds in Zhalajiao during the fermentation process. The study comprehensively examines changes in physical and chemical properties, microbial diversity, and flavor components at various fermentation stages. It successfully identifies six organic acids, 17 amino acids, and 21 key volatile compounds as crucial flavor components in Zhalajiao. Furthermore, it highlights the significant roles of Lactobacillus and Cyanobacterium in flavor formation, with Candida being the primary fungal genus associated with flavor development. These findings offer valuable insights into the regulation and quality assurance of Zhalajiao's flavor.

Specific Comments:

Please rephrase the sentence, "However, the formation mechanism of the crucial flavor components of Zhalajiao remains unclear" to read, "However, the mechanisms underlying the formation of these crucial flavor components in Zhalajiao remain unclear."

If feasible, consider adding information about the traditional fermentation duration in the following sentence: "Traditionally, it is produced through spontaneous fermentation by using cereals (mainly corn or rice) and fresh chilies (Xiaomila, Erjingtiao, and Niujiaojiao) as raw materials under the action of naturally occurring microorganisms."

The authors mentioned that the chilies were naturally fermented for 90 days. Please provide an explanation for why the authors chose this specific duration as the maximum fermentation period and whether there is any correlation with the traditional fermentation duration.

It is necessary to justify why the gas chromatography separation conditions were employed as described in reference [14], especially considering that a different material (with a potentially different volatile profile) was used in [14] (radish paocai).
